# Nanoscale-femtosecond dielectric response of Mott insulators captured by two-color near-field ultrafast electron microscopy

Xuewen Fu [1,2,7✉], Francesco Barantani [3,4,7], Simone Gargiulo [3,7], Ivan Madan[3], Gabriele Berruto[3], Thomas LaGrange[3], Lei Jin[5], Junqiao Wu [5], Giovanni Maria Vanacore [6], Fabrizio Carbone [3✉] & Yimei Zhu[2✉]

Characterizing and controlling the out-of-equilibrium state of nanostructured Mott insulators hold great promises for emerging quantum technologies while providing an exciting playground for investigating fundamental physics of strongly-correlated systems. Here, we use two-color near-field ultrafast electron microscopy to photo-induce the insulator-to-metal transition in a single $VO_2$ nanowire and probe the ensuing electronic dynamics with combined nanometer-femtosecond resolution ($10^{-21}$ m • s). We take advantage of a femtosecond temporal gating of the electron pulse mediated by an infrared laser pulse, and exploit the sensitivity of inelastic electron-light scattering to changes in the material dielectric function. By spatially mapping the near-field dynamics of an individual nanowire of $VO_2$, we observe that ultrafast photo-doping drives the system into a metallic state on a timescale of ~150 fs without yet perturbing the crystalline lattice. Due to the high versatility and sensitivity of the electron probe, our method would allow capturing the electronic dynamics of a wide range of nanoscale materials with ultimate spatiotemporal resolution.

[1] School of Physics, Ultrafast Electron Microscopy Laboratory, Nankai University, Tianjin 300071, China. [2] Condensed Matter Physics and Material Science Department, Brookhaven National Laboratory, Upton, NY 11973, USA. [3] Institute of Physics, Laboratory for Ultrafast Microscopy and Electron Scattering (LUMES), École Polytechnique Fédérale de Lausanne, Station 6, Lausanne 1015, Switzerland. [4] Department of Quantum Matter Physics, University of Geneva, 24 Quai Ernest-Ansermet, 1211 Geneva 4, Switzerland. [5] Department of Materials Science and Engineering, University of California, Berkeley, CA 94720, USA. [6] Department of Materials Science, University of Milano-Bicocca, Via Cozzi 55, 20121 Milano, Italy. [7] These authors contributed equally: Xuewen Fu, Francesco Barantani, Simone Gargiulo. ✉email: xwfu@nankai.edu.cn; fabrizio.carbone@epfl.ch; zhu@bnl.gov

The ability to investigate and actively control electronic, optical and structural properties of quantum materials is key to addressing the pressing demands for sustainable energy, high-speed communication/computation, and high-capacity data storage[1–3]. These research aims require the development of quantitative methods and techniques that enable to visualize and control the complex structural, electronic and dielectric evolution of nanomaterials at the proper temporal (femtosecond (fs)) and spatial (nanometer (nm)) scales. This is particularly relevant for the case of strongly correlated materials undergoing electronic–structural phase transitions, as, for instance, the representative of Mott systems, vanadium dioxide ($VO_2$), which undergoes an insulator-to-metal transition (IMT) slightly above room temperature (~340 K). $VO_2$ has recently attracted a renewed interest due to the promising applications in emerging technologies, such as volatile memories and neuromorphic computation[1,4,5]. This scenario becomes even more intriguing when the physical dimensions of the Mott systems shrink down to nm length scales. This is, in fact, the typical dimensions of the basic building blocks for most Mottronic devices, where quantum confinement and surface effects may lead to substantial modifications of the structural and electronic dynamics during the Mott transition process. Therefore, it is crucial to provide a deeper understanding of the out-of-equilibrium interplay between electronic and structural degrees of freedom in individual nanoscale Mott systems, which can be achieved only when both spatially and temporally resolved information is simultaneously retrieved.

So far, the underlying mechanism and structural dynamics of the IMT in the $VO_2$ have been intensively studied by ultrafast X-rays diffraction[6–8] and ultrafast electron diffraction[9–11], which are indeed able to investigate the material's behavior with combined fs/atomic-scale resolution and have been shown to provide crucial insights into the structural mechanisms governing the Mott transitions. However, these approaches are not able to give direct information on the electronic degrees of freedom and are mainly limited to bulk crystals, thin films, or clusters of many nanostructures. On the other hand, time-resolved optical pump–probe techniques, such as reflectivity, ellipsometry, and photoemission, can provide access to detailed information on the fs dynamics of the initial electronic processes in the phase transitions by measuring the dielectric response[12–14]. However, such methods are inherently limited to a spatial resolution of micrometer scale, because of the long wavelength of the optical probes. Spatial information at smaller length scales can usually be obtained using conventional electron microscopy techniques[15,16] or coherent x-ray imaging methods[17,18], which can investigate single nanostructures with high spatial resolution, but without temporal resolution. Recently, a photon gating approach in transmission electron microscopy has been demonstrated for studying the phase transition of $VO_2$ nanostructures[19], where the authors have investigated the spatially average IMT dynamics of an ensemble of $VO_2$ nanoparticles. Therefore, hitherto, investigation of the out-of-equilibrium interplay between electronic and structural degrees of freedom in $VO_2$ has been mainly devoted to bulk crystals, thin films or ensemble measurements, whereas it has been challenging on individual nanostructures for which there is a substantial deficiency of relevant experimental data.

Ultrafast electron microscopy (UEM) is the most promising choice to address such challenge due to its unique high spatio-temporal imaging capabilities[20–30]. The compelling aspect of the UEM technique is the wide range of information provided by the analysis of the transmitted electrons coupled with high temporal resolution not limited by the detector response. It is possible to perform real-space imaging at the nm scale, record diffraction patterns for structural and lattice dynamics investigations, or

acquire electron spectra with sub-eV resolution[20–30]. The latter case is generally referred to as electron energy loss spectroscopy (EELS) and provides a measure of the loss function of electrons scattered from the material ($\Im\{1/\varepsilon\}$). In an ideal case, from the knowledge of the loss function and using Kramers–Kronig relations, one can retrieve the complex dielectric function of the material under investigation[31]. As the dielectric function directly correlates with the electronic degrees of freedom in materials, it would be possible to access the electronic dynamics across the IMT in the Mott systems via measuring the dielectric function change. However, such approach works well only when the loss function is known over the whole energy range. In a real situation, such constraint is generally hard to achieve, especially in the low-energy region. Moreover, the typical temporal resolution of UEM experiments is on the order of several hundreds of fs even with a single electron per pulse in the absence of space charge effects[23,24,28–30]. This value is determined by the statistical distribution of the time of arrival of electrons photoemitted from the photocathode with slightly different velocities, and it is generally much larger than the typical timescales of the electronic motion in materials, which tend to be on a few tens of fs or below. Although "photon gating"[19,32] and several electron pulse compression schemes, such as microwave compression[33] and terahertz compression[34], have been proposed to improve the temporal resolution of UEM into the timescale of electronic motion, implementation of pump–probe nanoscale imaging of material electronic dynamics only lasting few tens of fs have not been experimentally observed yet.

Here, we overcame these issues by exploiting a well-established variant of the UEM technique named photon-induced near-field electron microscopy (PINEM). In PINEM, inelastic electron–light interaction takes place in the presence of nanostructures when the energy-momentum conservation condition is satisfied[35]. As a result of such interaction, electrons inelastically exchange multiple photon quanta that can be resolved in the electron energy spectrum as a series of discrete peaks, spectrally spaced by multiples of the photon energy ($n\hbar\omega$) on both sides of the zero-loss peak (ZLP)[35–37]. The largest electron–light interaction is achieved when the optical pulse and electron pulse arrive simultaneously at the specimen. As shown in the theory[38,39], the fundamental quantity to describe the PINEM process is the scattered field integral $\beta(x, y)$ defined as:

$$\beta(x,y) = \frac{e}{\hbar\omega} \int_{-\infty}^{+\infty} dz\, E_z(x,y,z) e^{-\frac{i\omega z}{v_e}}, \qquad (1)$$

where $E_z$ is the component of the electric field along $z$ (electrons propagation direction), $\omega$ the frequency of the laser, and $v_e$ the speed of electrons. The probability of interaction between the near-field and the electron is determined by:

$$P_n = [J_n(2|\beta(x,y)|)]^2, \qquad (2)$$

where $J_n$ is the Bessel function of the first kind of order $n$, and it represents the probability of exchanging $n$ quanta of light. The PINEM spectrum is then given by the sum over all the contributions:

$$I_{PINEM} = \sum_{|n|=1}^{\infty} [J_n(|\beta(x,y)|)]^2. \qquad (3)$$

In the case of weak interaction, one can write that $J_n(\rho) \sim \rho^n$, and thus the expression for the PINEM spectrum becomes:

$$I_{PINEM} \propto \sum_{|n|=1}^{\infty} |\beta(x,y)|^{2n}. \qquad (4)$$

For a cylindrical nanowire (NW) and within the weak interaction approximation the field integral $\beta$ can be approximated at

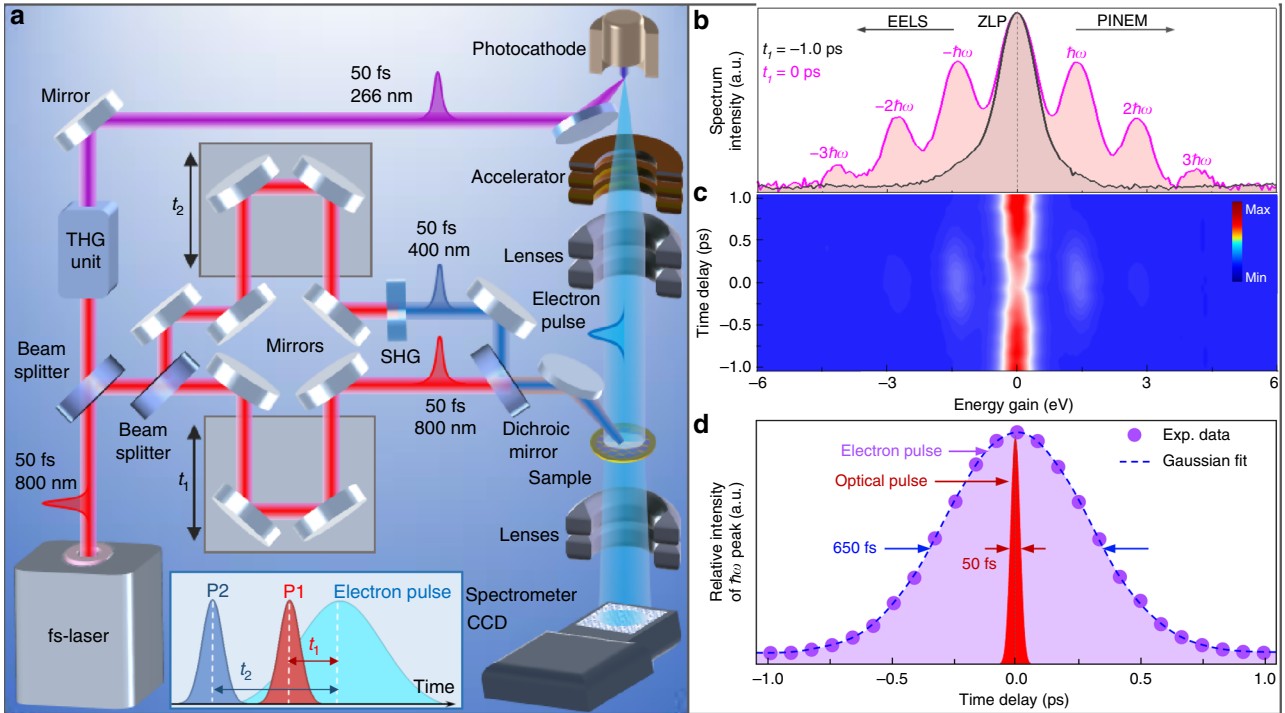

**Fig. 1 Two-color near-field electron microscopy. a** Experimental set-up for two-color near-field electron microscopy. NIR 50-fs optical pulses (800 nm) are separated into two parts by a beam-splitter. One portion is frequency-tripled to generate UV pulses (266 nm) via a third harmonic generator (THG) unit, which are then directed to the photocathode to generate ultrafast electron pulses. The other portion of the NIR laser is further split into two parts: one is left unchanged (800 nm, P1), while the other is frequency doubled by a second harmonic generator (SHG) to generate visible optical pulses (400 nm, P2). The time delays $t_1$ and $t_2$ between the two optical pulses and electron pulse are controlled by two linear delay stages. The visible and NIR pulses are recombined via a dichroic mirror and focused onto the specimen in the microscope. The bottom inset shows the time axis of the visible (P1) and NIR pulses (P2) and their delays ($t_1$, $t_2$) relative to the electron pulse at the specimen plane. **b, d** One-color PINEM results of a single $VO_2$ NW (diameter of ~350 nm) using the P1 optical pulse (fluence of ~4.1 mJ/cm²) with the field polarization perpendicular to the NW axis. **b** Two representative PINEM spectra at $t_1 = -1.0$ ps (gray curve) and 0 ps (pink curve), respectively, in which each spectrum is normalized by its own maximum intensity. **c** PINEM spectrogram of photon–electron coupling between the P1 pulse and the electron pulse as a function of the P1 delay time ($t_1$). **d** Cross-correlation temporal profile (violet dots and fitted blue curve) obtained by integrating the PINEM peaks at each instant of the P1 pulse arrival time ($t_1$); the plot shows the temporal profile of the electron pulse (~650 fs). The shaded red curve represents, instead, the time window of the optical gating, which corresponds to the temporal duration of the optical pulse P1 (50 fs).

first order as[40]:

$$\beta \approx \frac{e}{\hbar\omega} E_0 \cos\phi \chi_c a^2 \frac{\omega}{v_e} e^{-\frac{\omega}{v_e}b}, \quad (5)$$

where $\phi$ is the azimuth angle to the polarization direction, $\chi_c = \frac{2\varepsilon(\omega)-1}{\varepsilon(\omega)+1}$ is the cylindrical susceptibility, $a$ is the NW radius, and $b$ is the impact parameter representing the in-plane projected distance between the electron and the NW central axis[40,41].

From Eqs. (4) and (5), we note that the dielectric function $\varepsilon(\omega)$ at the optical pump frequency $\omega$ is directly encoded in the PINEM signal through the localized near-field generated around the nanostructure. As the lifetime of such near-field is defined by the optical pulse, which is considerably smaller than the electron pulse duration, PINEM is inherently a stationary method. To measure the temporal evolution of the dielectric response and the related electronic dynamics of materials, in addition to the infrared optical pulse (P1) that creates the PINEM, we have introduced an additional visible optical pulse (P2) to excite the sample transiently (Fig. 1a). P1 is synchronous with the electron pulse at the specimen to create PINEM for probe, while P2 is delayed with respect to P1 by a variable delay time for photon pump: a scheme that could be concisely referred to as two-color photon-pump/PINEM-probe experiment[19]. Such a method takes both advantages of the direct relation between localized near-field

intensity and dielectric function, and the strong enhancement of the localized near-field excitation. It can potentially retrieve $\varepsilon(\omega)$ with an energy resolution entirely determined by the laser bandwidth (about 20 meV) rather than the electron bandwidth (about 1 eV) as in the regular EELS case[42]. Another crucial advantage with respect to regular EELS or PINEM regards the temporal resolution. In our two-color PINEM approach, the probe is no longer the initial, primary photoelectron pulse, whose temporal duration is typically on the order of several hundreds of fs. The probe in this approach is actually given by the inelastically scattered electrons, whose temporal duration is instead determined by the infrared laser pulse (P1) that drives the electron–light coupling. Thus, the infrared pulse P1 acts as a 'temporal gate' of the electron pulse[19,32]. A two-color PINEM experiment, where such gated electrons are measured as a function of the delay time between the two optical pulses (P2 and P1), thus has a temporal resolution only determined by the laser pulse (50 fs in our case), which almost an order of magnitude better than a normal UEM experiment where the gate pulse P1 is absent.

To demonstrate the capabilities of our approach, we have investigated the photoinduced IMT occurring in a single $VO_2$ NW. The two-color PINEM imaging allowed to retrieve the dielectric response of the $VO_2$ NW with combined nm–fs resolutions. We reveal that ultrafast photo-doping drives the NW into a metallic state on a timescale of ~150 fs, as a result of a

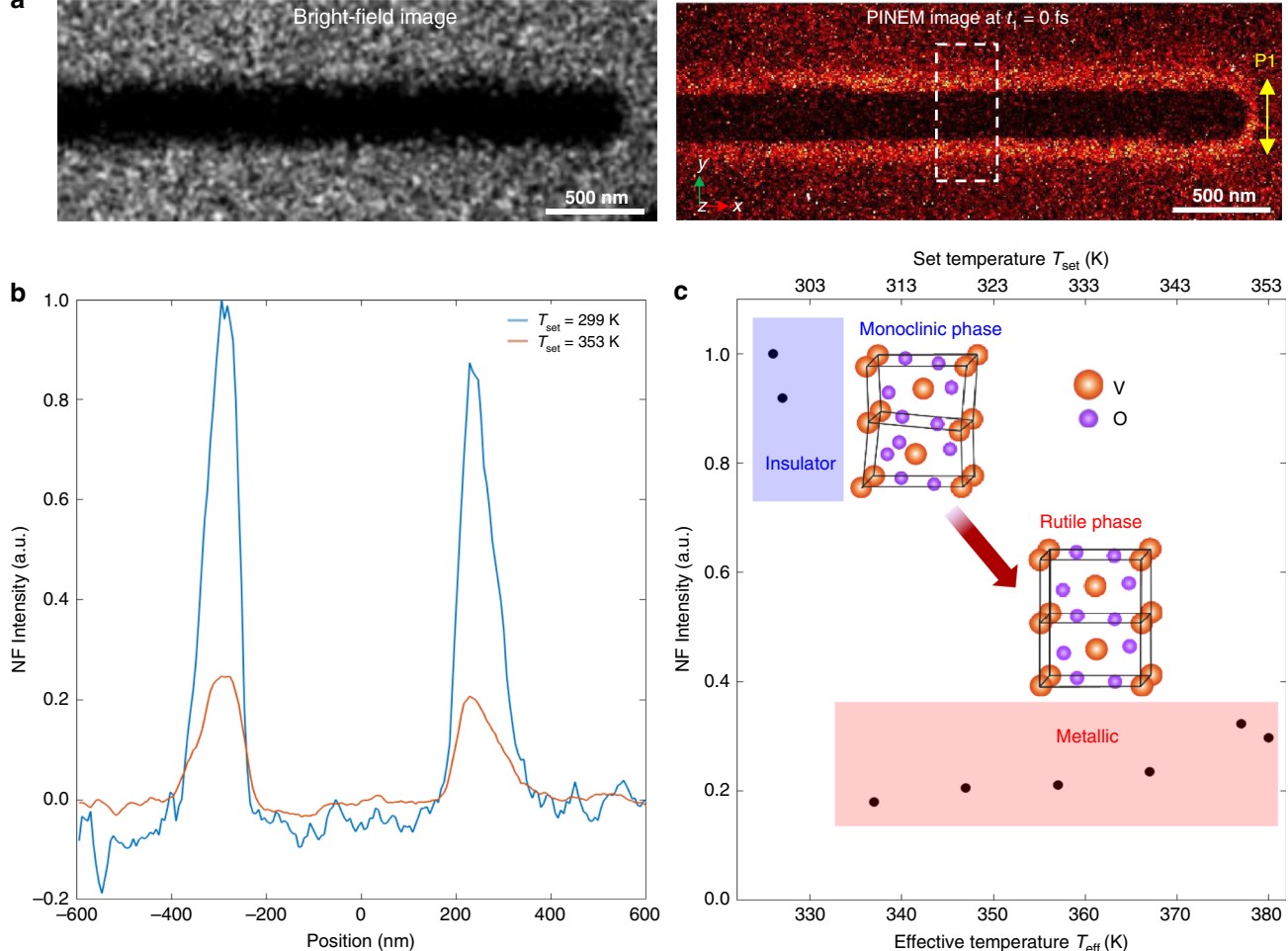

**Fig. 2 One-color PINEM experimental result of a single VO₂ NW at different temperatures. a** Left panel: bright-field image; right panel: typical energy-filtered PINEM image ($t_1 = 0$ fs) of the investigated VO₂ NW (~350 nm in diameter) with the P1 optical pulse polarized perpendicularly to the NW axis. **b** Spatial distribution of the PINEM signal across the NW at $T_{set} = 299$ K (blue line, below transition) and $T_{set} = 353$ K (orange line, above transition). The intensity was integrated along the NW axis in the area indicated by the dashed white box in the PINEM image shown in the right panel of **a**. The middle plane of the NW along the $y$ direction is defined as zero for the positions in the horizontal axis. For this one-color PINEM measurement, 800 nm optical pulses (fluence of ~4.1 mJ/cm²) were used and the time delay was set at $t_1 = 0$ fs to maximize the PINEM coupling. **c** Integrated PINEM intensity of the NW as a function of temperature. $T_{set}$ is the set temperature of the sample on the heating holder, and $T_{eff}$ is the effective temperature, which differs from the $T_{set}$ due to an additional temperature jump (~31 K) induced by the 800 nm optical pulse (P1). The PINEM intensity in the vicinity of the VO₂ NW significantly decreases when raising the temperature above the transition point (~340 K). The insets show the lattice structures of insulating monoclinic phase and metallic rutile phase, respectively.

photocarrier-driven change of the interatomic potential, without yet perturbing the crystalline lattice. This is then followed by an ensuing recovery to the electronic equilibrium on a tens of picosecond (ps) timescale related to the anharmonic excitation of transverse acoustic phonons. Such observations elucidate the crucial role of the electronic dynamics for initiating the IMT in the VO₂ NW.

## Results and discussion

**One-color PINEM of a single VO₂ NW.** First, we performed a one-color PINEM experiment on a single VO₂ NW to characterize the features of its PINEM spectrum at both insulating and metallic phases. The single-crystal VO₂ NWs were synthesized by chemical vapor deposition (see "Methods" section) and were directly transferred to an amorphous silicon nitride membrane (Si₃N₄, 20-nm thick) window for measurements. The left panel of Fig. 2a shows a bright-field image of the NW (diameter of ~350 nm). In this experiment, only the first optical pulse P1

(duration of 50 fs, $\lambda = 800$ m, fluence of ~4.1 mJ/cm²) was used to excite the sample with a polarization perpendicular to the NW axis to maximize the near-field excitation. Electron energy spectra were measured as a function of time delay ($t_1$) between the optical pulse P1 and the electron pulse (Fig. 1a). The recorded PINEM spectra are presented in Fig. 1c. Discrete peaks at integer multiples of $\hbar\omega$ appear on both sides of the ZLP (the latter is shown as a shaded area in Fig. 1b at $t_1 = 1.0$ ps and $-1.0$ ps), and exhibit a maximum intensity at $t_1 = 0$ fs (the shaded pink curve). As the P1 optical pulse is much shorter than the electron pulse, the temporal profile of the PINEM intensity shown in Fig. 1d is mainly determined by the electron pulse duration (~650 fs via a Gaussian fitting), which represents the typical temporal resolution of regular UEM experiments. As a result of the "photon gating" in a two-color PINEM experiment, the temporal duration of the inelastically scattered electrons (PINEM electrons) is instead on the order of ~50 fs (given by the gating optical pulse duration[19,32]), thus improving the temporal resolution of about one order of magnitude.

To quantitatively characterize the localized near-field and the dielectric function of the $VO_2$ NW in insulating and metallic phases, we acquired energy-filtered images while thermally heating the sample across the IMT. A real-space map of the localized near-field is retrieved by selecting only those electrons that have acquired photon energy quanta (see "Methods" section). In these measurements, the time delay between the P1 optical pulse and the electron pulse is fixed at $t_1 = 0$ fs to attain the maximum electron–photon coupling. Typical room temperature bright-field image (left panel) and energy-filtered PINEM image (right panel) are shown in Fig. 2a. The optically induced near-field appears as a bright-contrast region surrounding the $VO_2$ NW. The maximum PINEM coupling at $t_1 = 0$ fs can be also clearly verified by the temporal dependent PINEM images in Movie S1. Figure 2b shows two typical spatial distributions of the PINEM intensity across the NW (obtained by integration along the NW axis in the dashed white box indicated in the right panel of Fig. 2a) at two different set temperatures, $T_{set} = 299$ K (blue line) and 353 K (orange line), respectively. Note that the optical pulse P1 also induces an additional temperature jump on the NW (~31 K, see "Methods" section), and thus the corresponding effective temperatures are 330 K (below IMT) and 384 K (above IMT), respectively. As shown in Fig. 2b, when thermally heating the nanowire across the IMT temperature (340 K), the PINEM intensity shows a pronounced decrease. This effect is further supported in Fig. 2c, which depicts the temperature dependence of the integrated intensity. An abrupt drop is observed when crossing the transition temperature, where the NW transforms from the monoclinic insulating phase into the rutile metallic phase (see insets in Fig. 2c). Such behavior can be attributed to the sudden decrease of the $VO_2$ dielectric function at the photon energy 1.55 eV[43], typical of a first-order transition from the insulating to the metallic phase, which results in a substantially smaller susceptibility. Such weaker dielectric response is thus responsible for a weaker localized near-field, and thus a reduced PINEM coupling.

To further confirm the dielectric origin of the observed behavior, we have performed numerical simulations using a finite element method, where we calculate the scattered field from a single $VO_2$ NW illuminated by an 800-nm optical field (see "Methods" section). In the calculations, the incident field is chosen to be linearly polarized along the $y$-axis (perpendicular to the NW axis) and propagating along $z$ negative direction (Fig. 3a). The transition is modeled as a variation of the dielectric function from $\varepsilon_{ins} = 5.68 - i3.59$ ($\tilde{n}_{ins} = 2.49 + i0.72$) in the insulating phase to $\varepsilon_{met} = 2.38 - i3.26$ ($\tilde{n}_{met} = 1.79 + i0.91$) for the metallic phase as derived from ref. [43] for the case of a thin film. In our experiment, we consider that the dielectric function of a thin film might represent a good approximation of the dielectric environment of our $VO_2$ NWs, whose dielectric properties do not significantly differ from those of the bulk or high-quality thin films, especially for optical wavelengths in the visible range as adopted in our experiment[44]. The dielectric permittivity of the $Si_3N_4$ substrate is kept constant[45]. Also, all plasmonic effects for the investigated $VO_2$ nanowires are intrinsically taken into account within the finite element simulations.

Figure 3b represents the computed spatial distribution of the field integral $\beta$ projected on the NW plane ($xy$ plane) for the insulating and metallic cases, together with their difference map. In Fig. 3c, we plot instead the absolute value of $\beta$ for the two phases integrated along the $x$-direction and shown as a function of the spatial $y$-coordinate across the NW. The simulations clearly confirm that the reduced permittivity in the metallic phase is responsible for a weaker scattered field at the NW/vacuum interface, which results in the lower intensity of the PINEM signal as observed in our one-color experiment (Fig. 2b, c).

Such pronounced contrast in the PINEM signal and the qualitative agreement between experiments and simulations attest to the sensitivity of our approach to probe modifications of the dielectric function crossing the IMT and thus the ability to monitor its ultrafast dynamics with a two-color PINEM approach. Such high sensitivity of the technique is obviously not limited to the 800-nm light used here, but it extends to a wide range of light wavelengths.

**Two-color PINEM of a single $VO_2$ NW**. We present here the two-color PINEM experiment performed on a single $VO_2$ NW. As we will describe below, such a method allowed us to monitor the transient change of the dielectric function of the NW accessing the IMT dynamics with combined fs and nm resolutions. For a conceptually clean experiment (Fig. 1a), two essential requirements need to be fulfilled: (i) the pump optical pulse P2 driving the Mott transition must photo-excite the $VO_2$ NW into the metallic phase without producing any appreciable near-field (i.e., PINEM signal), and (ii) the optical gating pulse P1 must have sufficient fluence to produce PINEM with an intense near-field signal but below the threshold to trigger the Mott transition. In our case, an optical pulse (P2) with a central wavelength of 400 nm (duration of 50 fs) drives the transition, at which $VO_2$ exhibits a significant optical absorption. This P2 optical pulse was set at a fluence of ~15.3 mJ/cm² and was characterized by a polarization parallel to the NW axis, which minimizes the near-field excitation as the near-field only occurs at the end of the NW in such configuration (see the right panel of Fig. 2a). For the PINEM probe, the electron pulse was spatiotemporally coincident with a low fluence (~4.1 mJ/cm²) gating optical pulse (P1) with a central wavelength of 800 nm (duration of 50 fs). This P1 optical pulse was polarized along the direction perpendicular to the NW axis to maximize the optically induced near-field, and its time delay relative to the electron pulse was fixed at $t_1 = 0$ fs to maximize the PINEM coupling (see Movie S1).

In $VO_2$, a density-driven photoinduced IMT has been proven to occur with a critical energy dose $\Delta H_C$ of 2 eV/nm³[3,46]. For the case of a NW, the optical energy densities injected by the P1 and P2 pulses can be evaluated by resorting to scattering theory. In such framework, we determined the absorption cross section, $C_{abs}$, for the case of an infinite cylinder (see "Methods" section). The evaluated cross section is $2.63 \times 10^{-7}$ m in the case of 400 nm (P2), while at 800 nm (P1) $C_{abs}$ is $3.06 \times 10^{-7}$ m. We then computed the absorbed optical energy density (energy per unit volume) injected within the NW as:

$$\rho_{E,ph} = \frac{C_{abs}}{\pi a^2} \phi_{inc}, \quad (6)$$

where $\phi_{inc}$ is the incident fluence and $a$ is the cylinder radius. By considering the experimentally adopted parameters and geometrical configurations, we found that $\rho_{E,ph}$ is about 1 eV/nm³ at 800 nm, while increases to 3.5 eV/nm³ at 400 nm. The fact that $\rho_{E,ph}$ for 800 nm excitation is well below the critical energy dose $\Delta H_C$, whereas $\rho_{E,ph}$ for 400 nm is well above, is a strong confirmation that P2 is able to trigger the ultrafast IMT while P1 acts only as PINEM probe. Note that, electron beam may also induce effects on the IMT in $VO_2$, such as lowering the IMT temperature by creating oxygen vacancies[47]. However, the dose of the electron pulse in our experiment is several orders of magnitude smaller than the conventional thermal electron beam and its effect is negligible.

As mentioned above, the amplitude of the localized near-field created around the NW by the P1 optical pulse (800 nm) strongly depends on the susceptibility and thus on $\varepsilon(\omega)$. Besides the amplitude, also the decay length of such near-field can be directly connected to the material permittivity. In fact, according to Mie scattering theory for an infinite long cylinder the complex

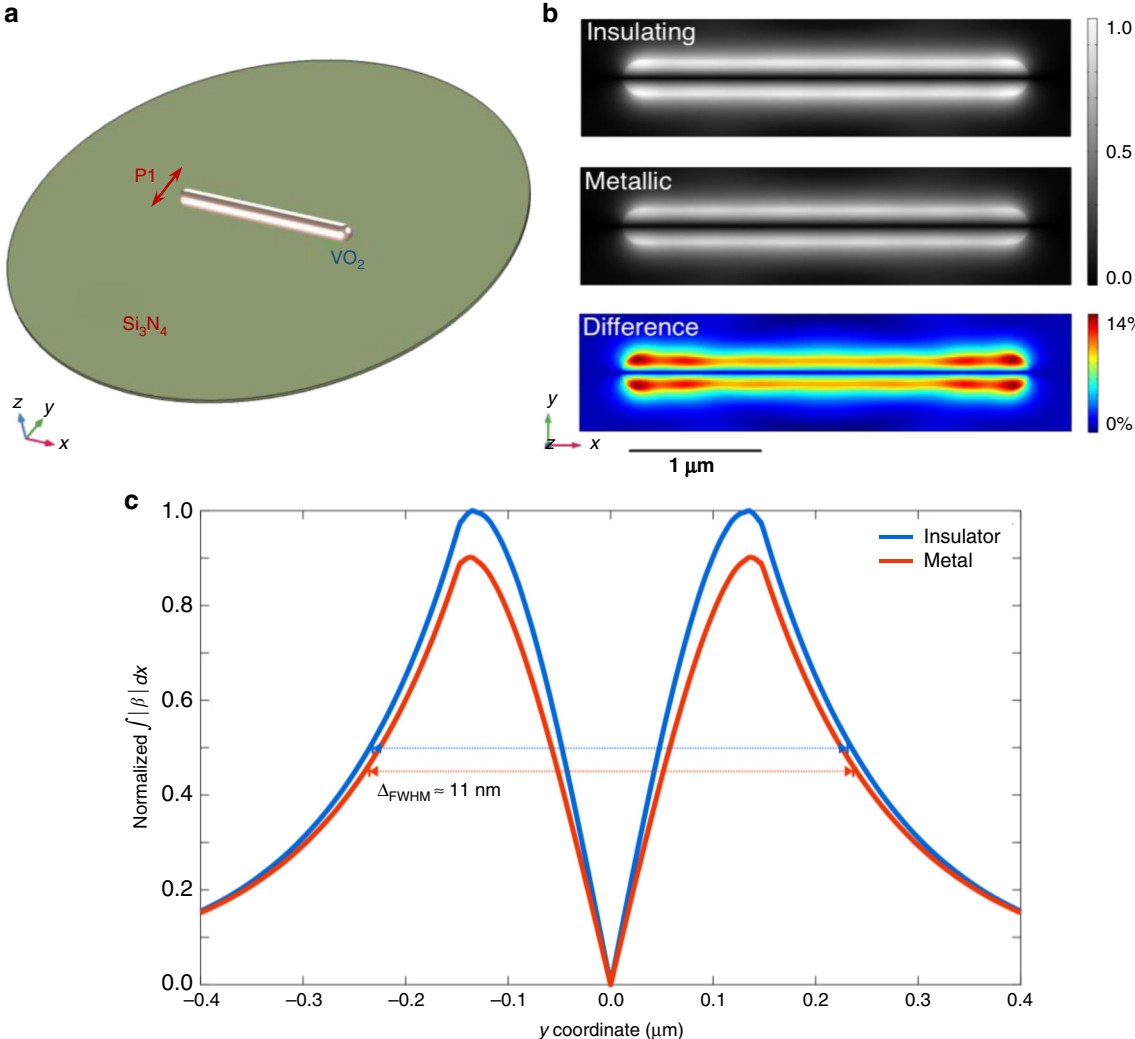

**Fig. 3 Numerical simulations of the one-color PINEM experiment. a** Simulation geometry of a 3 μm long VO$_2$ NW (300 nm diameter) placed on a 50 nm Si$_3$N$_4$ substrate. The red arrow shows the polarization direction of P1 optical pulse. **b** Simulated results of the interaction strength |β| as obtained from the scattered near-field distribution around the VO$_2$ NW for an insulating phase (top panel) and a metallic phase (middle panel), together with their difference map (bottom panel). **c** Simulated interaction strength integrated along the whole NW and plotted as a function of the spatial y-coordinate across the NW for both the insulating (blue line) and metallic (orange line) phases. Both curves are normalized to the peak maximum obtained in the insulating phase. The arrows represent the FWHM for the insulating (blue) and metallic (red) cases. Simulation reveals a difference of about 11 nm between the two widths, in favor of the metallic phase, as observed experimentally.

refractive index exhibits a direct spatial dependence[48]. This means that a change in the dielectric response induced by the P2 optical pulse (400 nm) can lead both to a modification of the PINEM intensity and, at the same time, to a different decay length of the PINEM signal at the NW/vacuum interface. To combine spatial and spectroscopic information, we acquired energy-filtered PINEM images at different delay times of $t_2$ (see "Methods" section). Figure 4a shows the bright-field image of the investigated VO$_2$ NW sitting on the Si$_3$N$_4$ membrane (left panel), and one of its corresponding energy-filtered PINEM image ($t_1 = 0$ ps, $t_2 = 0$ ps) (right panel), in which the transient near-field appears as a bright-contrast region surrounding it. To quantify the dielectric response to the IMT, we integrated the PINEM signal along the NW axis (within the area indicated by the blue box in the middle panel of Fig. 4a) and plotted it as a function of the position across the NW. Typical plots of such spatial distribution are shown in Fig. 4b for two different delay times, $t_2 = -0.4$ ps and $t_2 = 0.8$ ps ($t_1 = 0$ ps). For a more clear comparison of their lateral decay length, we also plot them in

Fig. 4c with each curve normalized to its own maximum. The PINEM intensity shows a substantial decrease at 0.8 ps, while the lateral decay length shows a simultaneous increase. The variation of the integrated PINEM intensity as a function of the delay time $t_2$ is plotted in Fig. 4d. Upon the 400 nm pump, the PINEM intensity shows an initial ultrafast decrease of ~30% with a time constant of ~155 fs, followed by a slower recovery on tens of ps timescale. Such behavior is consistent with a transformation to a metallic phase with a smaller permittivity, which results in a weaker PINEM coupling (consistently with the observations of one-color PINEM at high temperatures). Figure 4e shows the temporal evolution of the lateral width (measured as the full width at half maximum, FWHM) of the integrated PINEM signal surrounding the NW. We observed an ultrafast lateral increase of the localized near-field profile of about 40–60 nm with a time constant of ~240 fs, followed by a slower recovery on a ~10 ps timescale, whose dynamics is consistent with the intensity variation. Therefore, using our approach, it is also possible to follow the transition by looking at the change of the nanoscale

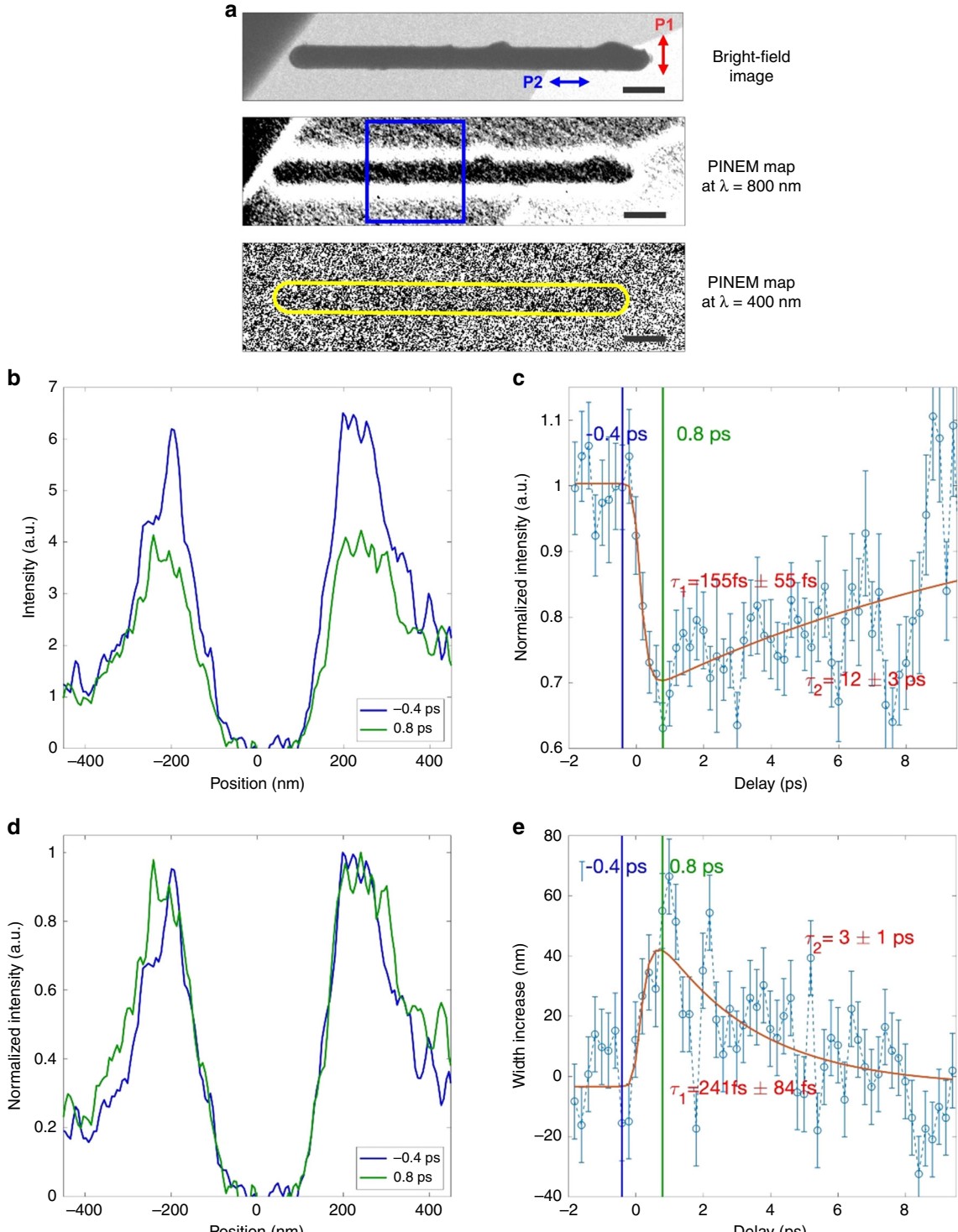

**Fig. 4 Two-color PINEM experimental result of a single VO$_2$ NW. a** Bright-field (top panel) and energy-filtered PINEM images measured at a pump wavelength of 800 nm (middle panel, $t_1 = 0$ ps) and at 400 nm (bottom panel, $t_2 = 0$ ps) of the investigated VO$_2$ NW (~350 nm in diameter) on the Si$_3$N$_4$ substrate. The red and blue arrows in the top panel show the polarization directions of P1 and P2 optical pulses, respectively. The blue box in the middle panel indicates the area along the NW where the PINEM signal is integrated. The scale bars are 500 nm. **b** Spatial distribution of the PINEM signal across the NW showed at the zero delay time $t_2 = -0.4$ ps (coincidence between pump and probe, blue curve) and 0.8 ps (green curve), respectively. **c** Normalized spatial distribution of the PINEM signal across the NW showed at the zero delay time $t_2 = -0.4$ ps (blue curve) and 0.8 ps (green curve), respectively. Both curves are normalized to their own maximum intensity. **d** Spatially integrated PINEM intensity as a function of delay time ($t_2$) between the 400 nm pump and the PINEM probe (open circles); the red curve is the best fit of the experimental data with a biexponential model convoluted with time-resolution of the technique. **e** Temporal dependence of the lateral spatial decay (measured at full width at half maximum) of the PINEM signal surrounding the NW. The red curve is the fit with two timescales convoluted with time-resolution of the technique. The error bars have been obtained by considering the measurement uncertainty and the variance within the counts.

spatial decay of the localized near-field. In both Fig. 4d, e, the red curves represent the best fit of the experimental data with a biexponential model where the temporal duration of the gated PINEM electrons and pump optical pulse P2 is explicitly taken into account. Importantly, it is also worth noting that the observed experimental behavior is well confirmed by the numerical simulations obtained for the two phases (see Fig. 2b), which show a lower interaction strength and a longer spatial decay in the metallic case with respect to the initial insulating state (see Fig. 3c).

**Microscopic mechanism for the IMT dynamics.** At a microscopic level, the Mott transition of bulk $VO_2$ from the monoclinic insulating phase to the rutile metallic phase proceeds through a series of transient steps characterized by a well-defined character and time constants[49]. The photoexcitation of the insulating phase corresponds to a photo-doping of the conduction band with excited electrons[50,51], and thus induces a redistribution of the electronic population within the 3d-symmetry bands. Such electronic rearrangement is thus responsible for a bandgap renormalization, which leads to an instantaneous collapse of the insulating bandgap (0.7 eV), rendering $VO_2$ metallic[50]. At the same time, the depopulation of bonding V-V orbitals is responsible for a strong modification of the double-well interatomic potential of the monoclinic lattice, which becomes highly anharmonic and flat. Ultrafast optical probes, which are particularly sensitive to the dielectric environment, have shown that such lattice potential change, which is related to a modified screening of the Coulomb interaction, occurs with a sub-100 fs time constant in bulk $VO_2$. At this point, the presence of a flat, anharmonic single-well potential triggers the atomic motions and leads to a removal of the long-range Peierls V-V dimerization. This has been observed with ultrafast structural probes to evolve on a 300–500 fs timescale[6,7,10]. Recently, large-amplitude uncorrelated atomic motions have also been noted to play a role in the V-V bond dilation on a timescale of about 150 fs[7]. Finally, the thermalization of the electronic population mediated by the excitation of transverse acoustic phonons over several ps temporal range can drive the lattice toward the final rutile metallic phase[9,10,49,51].

As shown in Fig. 4b, d, the localized PINEM signal from the $VO_2$ NW observed in our experiments exhibits an initial ultrafast dynamical process with a time constant of ~150 fs followed by a slower recovery process, which indicates the transition from the insulating to metallic phase without evidence of passing through any intermediate states. Furthermore, this value is nearly twice shorter than the ~300 fs of the coherent V-V displacement motions observed from previous structural probes[9,10], while it is consistent with the timescale for photo-doping-induced lattice potential change, which could abruptly unlock the V dimers and yield large-amplitude uncorrelated motions, as also observed from the ultrafast optical studies on the bulk crystals[7]. Thus, with our approach, we could capture the transient state in which the NW has become metallic as induced by the photocarrier-induced change of the interatomic potential, but before the crystalline lattice perturbations occur and so the structure remains in the monoclinic phase, namely, the transient quasi-rutile metallic phase[7]. Since both the PINEM intensity and its spatial distribution can be directly related to the dielectric function, our method is inherently sensitive to the electronic dynamics of the $VO_2$ NW. The advantage of our approach is the ability to retrieve such electronic dynamic information on a single nanostructure with combined nm–fs spatiotemporal resolution, which is particularly relevant, especially when nanoscale inhomogeneity plays a decisive role in the transition process[18].

Following the fs dynamics, we also observe a slower recovery toward the electronic equilibrium through the electron–lattice coupling on a timescale of tens of ps, which can be readily associated with anharmonic excitations of acoustic phonons. Thus, a themodynamically stable metallic rutile phase is not fully reached and then the system relaxes back to the insulating monoclinic phase.

It is worth noting that, because the photon-pump/PINEM-probe experiment was carried out in an ultrafast electron microscope, it would be possible to interrogate similar nanostructured materials under the same experimental conditions by ultrafast dark-field imaging or ultrafast diffraction using the temporally gated PINEM electrons. The latter would provide structural information with similar enhanced temporal and spatial resolutions and enable the possibility to simultaneously explore both the electronic (dielectric) and structural dynamics of the investigated individual nanostructures on a few tens of fs timescale.

In this work, we have implemented a two-color near-field UEM method and demonstrated its ability to access the initial ultrafast electronic process in the optically induced IMT in an individual $VO_2$ NW. We observed the temporal evolution of its dielectric response with PINEM imaging on nm and fs scales, achieving a combined spatiotemporal resolution several orders of magnitude larger than the conventional optical probes and static imaging. The high sensitivity of PINEM to the ultrafast dielectric response driven by laser photoexcitation attests to the high versatility of our approach for spatially resolved investigation of electronic dynamics and phase transitions that last a few tens of fs. Furthermore, incorporating with the advanced attosecond optical pulse generation techniques[32,52], it is feasible to achieve sub-fs and even attosecond temporal resolution in UEM via our approach. Therefore, this demonstration would be an important step towards the ultimate establishment of sub-fs/as resolution in electron microscopy for capturing electron motion in nanomaterials in real space and time.

## Methods

**Synthesis of $VO_2$ NWs.** The single-crystal $VO_2$ NWs were synthesized in a low-pressure horizontal quartz tube furnace by a chemical vapor transport deposition (CVD) method. In brief, $V_2O_5$ powder was placed in a quartz boat at the center of a horizontal quartz tube furnace, and a ~1.5 cm downstream unpolished quartz (~1 cm × 0.6 cm) was used as product collecting substrate. The furnace was heated to 950 °C to evaporate the $V_2O_5$ powder. Then the evaporated V-related species were transported by Ar carrier gas (6.8 sccm, 4 Torr) to the quartz substrate, and free-standing $VO_2$ NWs grew on the substrate surface. After 15 min, the CVD system naturally cooled down to room temperature.

**One- and two-color PINEM experiments.** A sketch of our one- and two-color PINEM experiments is depicted in Fig. 1a. The UEM used in this work, which is detailed in ref. [21], is a modified JEOL 2100 TEM integrated with an ultrafast amplified laser system delivering 50-fs pulses at 50 kHz repetition rate. The UV pulse (266 nm) impinges on the cathode photo-emitting electron pulses, while the 400 nm (P2) and 800 nm (P1) optical pulses (duration of 50 fs) are directly focused on the sample. The TEM is equipped with a Gatan imaging filter spectrometer, which allows us to acquire an energy-filtered image with a tuneable energy window. For the images presented in this work, we used a 15 eV wide energy window centered in the gain part of the spectrum with exposure time for the CCD sensor of about 60–90 s. For the one-color PINEM experiment, only use the 800 nm (P1) optical pulses to excite the specimen, with the optical polarization perpendicular to the NW axis to maximize the localized near-field excitation and PINEM coupling. For the two-color PINEM experiment, both 400 nm (P2) and 800 nm (P1) optical pulses were used to illuminate the specimen: the 800 nm (P1) optical pulse is linearly polarized perpendicular to the NW axis and fixed at the zero-time delay ($t_1 = 0$ fs) relative to the electron pulse to maximize the localized near-field excitation and PINEM coupling, whereas the 400 nm (P2) optical pulse for pump is linearly polarized parallel to the NW axis to minimize the localized near-field excitation and PINEM coupling.

In our experiments, we have recorded four scans in imaging mode and two scans in spectroscopy mode. They are all showing very similar results and therefore in the manuscript, we plot a representative dataset, which has been obtained by properly averaging the near-field around the NW.

**Scattering theory for an infinite cylinder**. To determine the absorption cross section for an infinite cylinder, we have resorted to scattering theory. In particular, we have used the derivation provided by Bohren & Huffman[48], where the scattering and extinction efficiencies in case of parallel (I) polarization (used for P2) are given by:

$$Q_{sca,I} = \frac{P_{abs,I}}{2aL\,\phi_{inc}} = \frac{2}{x}\left[|b_{0I}|^2 + 2\sum_{n=1}^{+\infty}\left(|b_{nI}|^2 + |a_{nI}|^2\right)\right], \quad (7A)$$

$$Q_{ext,I} = \frac{P_{ext,I}}{2aL\,\phi_{inc}} = \frac{2}{x}Re\left\{b_{0I} + 2\sum_{n=1}^{+\infty}b_{nI}\right\}. \quad (7B)$$

whereas in the case of normal (II) polarization (used for P1) we have:

$$Q_{sca,II} = \frac{P_{abs,II}}{2aL\,\phi_{inc}} = \frac{2}{x}\left[|a_{0II}|^2 + 2\sum_{n=1}^{+\infty}\left(|a_{nII}|^2 + |b_{nII}|^2\right)\right], \quad (7C)$$

$$Q_{ext,II} = \frac{P_{ext,II}}{2aL\,\phi_{inc}} = \frac{2}{x}Re\left\{a_{0II} + 2\sum_{n=1}^{+\infty}a_{nII}\right\}. \quad (7D)$$

Thus, the absorption efficiencies can be evaluated as:

$$Q_{abs,I} = Q_{ext,I} - Q_{sca,I} \cdot Q_{abs,II} = Q_{ext,II} - Q_{sca,II}, \quad (8)$$

where $a$ is the NW radius, $L$ its length, $a_n$ and $b_n$ are coefficients that depend on the Hankel and Bessel functions, as reported in Bohren & Huffman[53]. The absorption cross section per unit length $C_{abs}$ is related to the corresponding efficiency via the geometrical factor $2a$ as $C_{abs} = 2a\,Q_{abs}$. The evaluated cross section $C_{abs}$ is $2.63 \times 10^{-7}$ m in the case of 400 nm, while at 800 nm $C_{abs}$ is $3.06 \times 10^{-7}$ m. We then computed the absorbed optical energy density (energy per unit volume) injected within the nanowire as:

$$\rho_{E,ph} = \frac{C_{abs}}{\pi a^2}\phi_{inc}, \quad (9)$$

where $\phi_{inc}$ is the incident laser fluence. The numerical evaluation has been implemented using two different codes: MatScat[54,55] and ICOTOOL[56], which provided identical results. Here, we have considered $ñ_{ins} = 2.49 + i0.72$ at 800 nm and $ñ_{ins} = 2.09 + i1.31$ at 400 nm. By considering the experimentally adopted parameters and geometrical configuration, we found that the energy density injected at 800 nm is about 1 eV/nm³, while at 400 nm we have about 3.5 eV/nm³. The fact that $\rho_{E,ph}$ for 800 nm excitation is well below the critical energy dose $\Delta H_C$ for a photoinduced IMT phase transition of 2 eV/nm³, whereas $\rho_{E,ph}$ for 400 nm is well above, is a strong confirmation that only the blue light pulse will be able to drive the transition.

**Equilibrium heating model**. For the static heating experiment, it is important to estimate the contribution to the nanowire equilibrium temperature from the 800-nm light pulse. In first approximation, such temperature change can be obtained following a phononic heat capacity approach[57]. In this model, the absorbed flux, $f_{abs}$, is given by:

$$f_{abs} = \int_{T_0}^{T_0+\Delta T} C_{ph}(T)\,dT, \quad (10)$$

where $C_{ph}$ is the phononic heat capacity. The expression for the absorbed flux depends on the geometry of the structure. For the planar $Si_3N_4$ membrane, we can consider:

$$f_{abs}^{planar} = \alpha\phi_{inc}\left(\frac{h\nu - E_g}{h\nu}\right)(1 - R), \quad (11)$$

where $\alpha$ is the absorption coefficient, $h\nu$ the photon energy, $E_g$ the energy gap, $R$ the reflectivity, and $\phi_{inc}$ the laser fluence. For the case of a nanowire modeled by an infinite cylinder, the absorbed flux is instead given by:

$$f_{abs}^{cylinder} = \frac{C_{abs}}{\pi a^2}\phi_{inc}\left(\frac{h\nu - E_g}{h\nu}\right), \quad (12)$$

where $C_{abs}$ is the absorption cross section and $a$ is the cylinder radius. The heat capacity can be written as:

$$C_{ph}(T) = 9 n_a k_B \left(\frac{T}{\theta_D}\right)^3 \int_0^{\frac{\theta_D}{T}} \frac{x^4}{(e^x - 1)(1 - e^{-x})}\,dx, \quad (13)$$

where $n_a$ is the atomic density, and $\theta_D$ is the Debye temperature. When solving the integral equation Eq. (10) for the $VO_2$ NW using Eq. (12) for $f_{abs}$ and with the parameters specified in Table 1[43–45,58], we find that the temperature jump associated with the 800-nm infrared pulse is 31 K, thus below the critical transition temperature change (~60 K) and consistent with the observation of the thermally induced IMT. For the planar $Si_3N_4$ substrate, following the same approach although using Eq. (11) for $f_{abs}$, the temperature variation induced by the laser is smaller than 1 K and thus negligible.

**Numerical simulations**. The experimental geometry has been replicated to perform a finite element method simulation using COMSOL multiphysics. The 3D model is composed of the $VO_2$ NW and a thin $Si_3N_4$ substrate. In the outer part, a perfect matching layer (PML) was added to confine the solution. The $Si_3N_4$ substrate passes through the PML and can be considered infinite. The incident light wave is chosen to be linearly polarized along the $y$-axis (perpendicular to the NW axis) and propagating along $z$ negative direction. A parametric sweep of the real and imaginary part of the

**Table 1 Parameters used to estimate the temperature change.**

| Parameters | P1 (800 nm) |
|---|---|
| $Eg_{VO_2}$ | 0.7 eV |
| $Eg_{Si_3N_4}$ | 5.1 eV |
| $n_a$ $_{VO_2}$ | $9.68 \times 10^{28}$ atoms/m³ |
| $n_a$ $_{Si_3N_4}$ | $8.61 \times 10^{28}$ atoms/m³ |
| $\phi_{inc}$ | 4.1 mJ/cm² |
| $C_{abs}[VO_2]$ | $3.06 \times 10^{-7}$ m |
| $\alpha$ $_{Si_3N_4}$ | $1 \times 10^4$ m⁻¹ |
| $\theta_D^{VO_2}$ | 750 K |
| $\theta_D^{Si_3N_4}$ | 804 K |
| $a$ | 150 nm |
| $(1 - R)$ $_{Si_3N_4}$ | 0.90 |
| $\Delta T_{VO_2}$ | 31 K |
| $\Delta T_{Si_3N_4}$ | 0.19 K |

$VO_2$ complex refractive index ñ, using the values reported in ref. [43], replicates the insulator-to-metal transition. The dielectric permittivity of the $Si_3N_4$ layer remains constant since it exhibits a negligible variation over a temperature range of several hundreds of °C[45]. Moreover, the temperature change induced by the pump optical pulse on the $Si_3N_4$ membrane is much smaller than 1 K. The study is solved at a frequency of $\nu = 374$ THz (1.55 eV, 800 nm), considering a NW with dimensions 3 μm as length and 300 nm as diameter, placed on a 20 nm thick $Si_3N_4$ substrate. By taking advantage of the symmetry of both geometry and field source, the simulation volume has been reduced to one quarter, placing PMC and PEC surfaces in the $yz$ and $xz$ planes, respectively. The PINEM field $\beta$ and its spatial integral has been performed over a 3-μm region along $z$ and of 4 μm along $x$, i.e., a smaller region compared to the actual size of the simulation domain in order to minimize possible residual reflections from the boundary conditions.

Supplementary Movie S1: Temporal evolution of one-color PINEM images of a single $VO_2$ NW with P1 optical pulse (duration of 50 fs, $\lambda = 800$ m, fluence of ~4.1 mJ/cm²) polarized perpendicularly to the NW axis.

## Data availability
The data that support the findings of this study are available from the corresponding authors upon reasonable request.

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

## Acknowledgements

This work was supported by the Materials Science and Engineering Divisions, Office of Basic Energy Sciences of the U.S. Department of Energy under contract no. DESC0012704, and the National Nature Science Foundation of China (NSFC) at grant no. 11974191. The LUMES laboratory acknowledges support from the NCCR MUST and ERC Consolidator Grant ISCQuM. G.M.V. is partially supported by the EPFL-Fellows-MSCA international fellowship (grant agreement no. 665667). F.B. was supported by the Swiss National Science Foundation (Project no. 200020-179157). The materials synthesis was supported by U.S. NSF grant no. DMR-1608899.

## Author contributions

X.F. and Y.Z. conceived the research project. X.F., F.B., S.G., I.M., G.B. and G.M.V. did the experimental measurements and data analysis. L.J. and J.W. synthesized the sample. X.F., F.B., S.G., I.M., and G.M.V. wrote the manuscript with input from Y.Z., F.C., J.W., and T.L.G.. S.G. developed the model and performed the numerical simulations. All the authors contributed to the discussion and the revision of the manuscript.

## Competing interests

The authors declare no competing interests.
