## [Peer Review File · Nature Communications]

REVIEWER COMMENTS

Reviewer #1 (Remarks to the Author):

The characterization and control of nonequilibrium states of Mott nanostructured insulators are important for the development of new quantum technologies, providing also an experimental platform for studying the fundamental physics of strongly correlated systems. In the manuscript NCOMMS-20-22938, two-color ultrafast near-field electron microscopy was used to detect the laser-induced dielectric-metal transition in a single VO₂ nanowire and to observe the electronic dynamics in this sample. The femtosecond time gating of electronic pulses, mediated by an infrared laser pulse in the PINEM mode, provided ~ 100 fs time resolution in the experiment (Fig. 1 of the manuscript NCOMMS-20-22938). In the developed method, inelastic electron scattering is sensitive to a change in the dielectric function of the material. By spatial mapping of the near-field dynamics of a single VO₂ nanostructure, it was found that ultrafast laser-induced doping transformed the quantum system into a metallic state on a time scale of ~ 150 fs (in this case, the crystal lattice of the sample did not yet have time to rebuild).

It is worth noting that, since the experiment with laser excitation and probing in the PINEM mode was carried out in an ultrafast electron microscope chamber [NCOMMS-20-22938], in the future it will be possible on this basis to study similar nanostructured materials using ultrafast visualization in dark field mode or by ultrafast electron diffraction in PINEM geometry. The latter approach is able to provide not only structural information about the sample with high spatiotemporal resolution, but also provides the opportunity to simultaneously study both the electronic and structural dynamics of single nanostructures on a femtosecond time scale. Due to the high sensitivity of UED, such a method in the long run will allow detecting the electronic dynamics of a wide class of nanoscale materials with high spatial-temporal resolution.

COMMENTS:

1. Figure 1 in the manuscript NCOMMS-20-22938 is indicated as an illustration of the 2-color approach, however, the bulk of the information presented here relates to the 1-color mode. Please, pay attention to the inconvenient presentation of information in the right panel. Can the right panel be divided into several separate drawings? In this figure, are the designations t and t_1 identical?
2. What is delayed in Figures 2 - 4 along the horizontal axis? Why do the positions of the zero point and the designations of the horizontal axes differ?
3. In Figure 4, the designation is almost unreadable in the upper right panel.
4. The experiment was conducted with high-power laser radiation at a laser pulse repetition rate of 50 kHz. Did the authors verify that such a high repetition rate was not accompanied by undesirable thermal heating of the sample?
5. The V(IV) valence state disproportionate to V(III) and V(V). Control can be done by XPS technique. Please, add results of the analysis of the VO₂ sample before and after experiments.

Reviewer #2 (Remarks to the Author):

The authors introduce a novel technique for ultrafast near-field electron microscopy experiments on a single VO₂ nanowire to probe the photoinduced metal-insulator transition in a single nanostructure. As mentioned in the paper, this photoinduced transition has been extensively studied over the years through time-resolved electron scattering, probing structural dynamics, and ultrafast optical spectroscopies which typically measure the spatially averaged electronic response. Here the authors apply PINEM, photo-induced near-field electron microscopy, to measure the dynamics of the local

dielectric response of VO₂ following photoexcitation sufficient to induce an insulator to metal phase transition (IMT). In PINEM, electrons do not actually pass through the material but instead sense the local fields surrounding the structure via their contributions to the inelastic free-free transitions of an electron in a photon field.

The novelty of this work lies in the introduction of a second optical pulse to photoexcite the sample at a different wavelength and crossed polarization so that the interaction can be approximated as arising from two independent fields. Thus one can induce a material change with the first pulse and use the second optical pulse to gate the temporal response of the electron-matter interaction. In this way, the time-resolution of the technique can be less than the electron pulse duration.

In my opinion, this is really nice work showing the power of such a pump-probe PINEM technique to probe insulator to metal transitions in correlated electron materials, of which VO₂ is the quintessential example. I find the results of these experiments compelling and demonstrate that it is possible to spatially probe the dynamics of the photo-induced transition. This technique will undoubtedly be useful to study a variety of correlated electron systems yielding a photoinduced IMT where the local dielectric environment is drastically changed due to optical perturbation. In terms of learning new things about the material, I am not sure we really learn something new about VO₂ and they restrict their measurements to the low carrier density regime where the transition is of typical Mott character. It would have been interesting to drive the transition further above the threshold where a quasi-stable structural transition occurs from a monoclinic insulator to a rutile metal.

My first question is: why did the authors not do this? Surely they had enough optical fluence to push the system into the rutile phase. How close were they in doing this, by their estimates? The answer to this may be important in understanding the limitations of this technique. Perhaps the structural transition does not yield a sufficient dielectric change compared to the contributions of the mobile carriers. In this case, I am not certain I would agree that PINEM could be used to simultaneously probe structural dynamics.

If the authors have done a fluence dependence, does the long time constant they observe decrease with increasing fluence as reported in other studies?

The pump energy is higher in this study (3.1 eV) than the gating field (1.55 eV). For 800 nm excitation, one would expect an instantaneous metallization of Mott character followed by a rapid recovery of the insulating phase on a sub-picosecond time scale. Did the authors think to switch the role of the pump and gating pulses to verify this, or was there a technical reason why this could not be done?

Since the readout is mediated through the dielectric response of a nanostructure, and the analysis of this requires numerical modelling of the local near-field, it is necessary that the real and imaginary parts of the dielectric function are input for the gating field. The authors quote Ref. 43 however this work focuses on thin films of VO₂ which through the transition are composed of domains of conducting and insulating phases. How sure are the authors that the same dielectric functions appropriate for a film are applicable to a nanostructure where it is likely to be composed of one or two domains?

Are plasmonic effects negligible or are they taken into account through the finite element simulations?

The calculation of the carrier density in Equation 6 is likely too simplistic, given it uses things like a reflection coefficient for a nanostructure. Perhaps a scattering model is more appropriate to estimate the carrier density. An accurate number for the carrier density would be helpful to evaluate whether they are truly close to the structural transition.

On page 13, the authors assert that they see no evidence for an intermediate state on traversing the insulating to metallic phase. They indicate that previous works provide evidence for such an intermediate state, and they cite the work of Baum (Ref. 9) and Siwick (Ref. 10). While Baum proposed a two-step approach to the transition in 2007, the Siwick group to my knowledge has never made this assertion and in fact their data disproved it in 2014. They describe the transition as a disordering transition due to the non-thermal melting of the V-V dimerization in a fluence-dependent fraction of crystallites. They are quoted in their Ref. 10 work: "Thus, the fast dynamics correspond to nonthermal melting of the PLD in a fluence-dependent fraction of crystallites. In these crystallites, the vanadium atomic positions relax to their equilibrium R-phase separation on a time scale of 300 fs (26)." This was further explored in a recent PNAS paper (M. R. Otto et al., 116, 450 (2019)) in a percolative picture of the transition over the film, so this appears to be fairly consistent in their interpretation. No soft-mode phonons indicating an intermediate was seen in either work. This mistake can be forgiven as it appears to be a misinterpretation of the Siwick by the Wall Science paper (Ref. 7 in this manuscript) who appear to cite Siwick and Baum together concerning the intermediate. This should be likely be corrected in the literature but at the very least should not be propagated.

Overall, I believe this work to be highly interesting to the community and can provide useful information concerning correlated electron physics in nanostructures. I support publication of this work provided answers to some of the questions I've asked are reasonably answered and corrections are made.

Minor questions/comments:

In Fig. 4b there is a blue box drawn, however there is no obvious indication as to what it is.

For Fig. 4e, the fit to the width is allowed to go negative. What does this mean?

Reviewer #3 (Remarks to the Author):

Date: June 20, 2020

Re: Nature Communications manuscript NCOMMS-20-22938

The Fu et.al. manuscript is presenting the ultrafast dynamics of insulator-metal phase transition of single VO₂ nanowire by probing the photoinduced dielectric constant using the photon gating approach in the ultrafast electron microscope. This approach has been utilized to study the phase transition in VO₂ nanoparticles previously in ref#31 in the manuscript. The author should cite the previously mentioned reference in the introduction (i.e. line 54 & line 61). Although, the novelty of the presented work is the connection between the ultrafast dynamics and the morphology of single VO₂ nanowire which consider an important step forward in this approach and adding an impact in the field to study and image the ultrafast dynamics of solid-state by the ultrafast electron microscope.

The organization of the experiments and starting with the one-color PINEM experiment first to prove the effect of the phase transition on the PINEM spectrum is well presented.

The two-color PINEM experiment explanation and satisfaction of the two mandatory preconditions of this type of experiment, namely;

- 1- The pump optical pulse should not produce any PINEM signal, and
- 2- The optical gating pulse must be below the threshold to trigger the dynamics under study,

reflects the capability and understanding of the photon-gating PINEM experiment.

However, I think the authors miss present the satisfactory of condition #2 in figure 2 an (as the author refers to in line 222) where the middle and the left panel show the PINEM signal of PI and I think the author meant to present in the right panel that no obvious PINEM signal is generated from P2 pulse.

The author presented the change in the PINEM spectra due to the photoinduced phase transition and the change in the dielectric of VO₂ nanowire in Figure 4 a and then show the normalized spectra in Figure 4b. I do not think this necessary and it would confuse the reader. It should be discarded or move to supplementary information. The same applied to Figure 4 e.

It is important to improve the presentation quality of the results shown in figure 4d to show the error bar of the data and mention the numbered of the scans that have been recorded in the text.

Finally, in the conclusion section, the authors mentioned that the photon gating approach in electron microscopy would lead to achieving the attosecond resolution in electron microscopy by using the advances of in attosecond optical pulse generation then cite ref# 55& 56. Obviously, the authors are not from the attosecond physics field, and this probably why they cited these two references referring to optical attosecond pulses. These two citations reporting the advancement of XUV attosecond pulses which is totally different than optical attosecond pulse. Using the term "optical" means in the visible and nearby frequency ranges not extreme ultraviolet (i.e. XUV). The generation of PINEM inside the microscope using attosecond XUV pulses is technically very changeable due to the limitation of performing the HHG process (the only valid way so far in the field. to generate XUV attosecond pulses) inside the microscope. Moreover, even if this limitation is overcome (for example by considering the HHG process in solid), the XUV generation efficiency and flux are not sufficient to generate any PINEM signal. The PINEM-like experiment is reported in the attosecond physics field for free electrons two decades ago and it is called attosecond XUV streaking. Hence, in line (326) in the manuscript conclusion, citing reference#32 in the manuscript and [J. Phys. B: At. Mol. Opt. Phys. 51, 032005 (2018)] will be more accurate and realistic.

Considering the mentioned points is crucial for improving the work presentation. The manuscript figures (i.e. Fig 2 and Fig 4), language, and citations need to be corrected and revised.

Mohammed Th. Hassan
Assistant Professor of Physics and Optical Sciences
Physics Department and James C. Wyant College of Optical Sciences
University of Arizona
Attomicroscopy and Attosecond Electron Imaging group leader
Mail: 1118 E 4th Street, PAS 363,
Tucson, AZ. 85721-0081
Office: +1-520-626-1435
e-mail: mohammedhassan@email.arizona.edu
Webpage: <http://hassan.lab.arizona.edu>

Response to reviewers' comments

We would like to thank all three reviewers for their positive and constructive reviews, and for their valuable comments to our work, which have tremendously helped us to improve this manuscript. We have carefully addressed all the comments and concerns, point-by-point in this revised version. Our detailed responses are listed below.

Reviewer 1

Comments:

The characterization and control of nonequilibrium states of Mott nanostructured insulators are important for the development of new quantum technologies, providing also an experimental platform for studying the fundamental physics of strongly correlated systems. In the manuscript NCOMMS-20-22938, two-color ultrafast near-field electron microscopy was used to detect the laser-induced dielectric-metal transition in a single VO₂ nanowire and to observe the electronic dynamics in this sample. The femtosecond time gating of electronic pulses, mediated by an infrared laser pulse in the PINEM mode, provided ~ 100 fs time resolution in the experiment (Fig. 1 of the manuscript NCOMMS-20-22938). In the developed method, inelastic electron scattering is sensitive to a change in the dielectric function of the material. By spatial mapping of the near-field dynamics of a single VO₂ nanostructure, it was found that ultrafast laser-induced doping transformed the quantum system into a metallic state on a time scale of ~ 150 fs (in this case, the crystal lattice of the sample did not yet have time to rebuild).

It is worth noting that, since the experiment with laser excitation and probing in the PINEM mode was carried out in an ultrafast electron microscope chamber [NCOMMS-20-22938], in the future it will be possible on this basis to study similar nanostructured materials using ultrafast visualization in dark field mode or by ultrafast electron diffraction in PINEM geometry. The latter approach is able to provide not only structural information about the sample with high spatiotemporal resolution, but also provides the opportunity to simultaneously study both the electronic and structural dynamics of single nanostructures on a femtosecond time scale. Due to the high sensitivity of UED, such a method in the long run will

allow detecting the electronic dynamics of a wide class of nanoscale materials with high spatial-temporal resolution.

Response:

We thank the reviewer for her/his careful review and appreciate her/his evaluation on the promising potential of our approach for simultaneously studying both electronic and structural dynamics of single nanostructures with high spatial and temporal resolution.

Comments #1:

Figure 1 in the manuscript NCOMMS-20-22938 is indicated as an illustration of the 2-color approach, however, the bulk of the information presented here relates to the 1-color mode. Please, pay attention to the inconvenient presentation of information in the right panel. Can the right panel be divided into several separate drawings? In this figure, are the designations t and t_1 identical?

Response:

According to the referee’s suggestion, we have reshaped the right panel of Fig. 1 to make a more readable presentation of the information (see also page 28 of the revised manuscript). The designation t means t_1 . To avoid confusion, we have changed t to t_1 in the revised Fig. 1.

Revised version of Figure 1

Comments #2:

What is delayed in Figures 2 - 4 along the horizontal axis? Why do the positions of the zero point and the designations of the horizontal axes differ?

Response:

We thank the referee for this question. The horizontal axis displayed in figures 2-4 indicates the positions across the nanowire. To improve the presentation and avoid any confusion, we have defined the neutral plane of the nanowire as the zero point in all figures in the revised manuscript and figure caption (page 30), as also shown below.

“The middle plane of the NW along the y direction is defined as zero for the positions in the horizontal axis.”

Comments #3:

In Figure 4, the designation is almost unreadable in the upper right panel.

Response:

We have modified the designation in Fig. 4a to make the information clear in the revised manuscript (page 32).

Comments #4:

The experiment was conducted with high-power laser radiation at a laser pulse repetition rate of 50 kHz. Did the authors verify that such a high repetition rate was not accompanied by undesirable thermal heating of the sample?

Response:

We agree with the referee that the choice of appropriate repetition rate is important for ultrafast pump probe experiment, as one of the pre-conditions for pump-probe measurement is that the excited system should fully recover to the initial state before the next pump pulse arrives. The pump laser pulse excitation will likely induce a transient heating and temperature jump of the sample, which has also been discussed in the main text. The thermal dissipation process is known to be in the order of hundreds of ns to microseconds from optical experiments (see for instance, *Progress in Surface Science* **90**, 464 (2015)). The time interval between the adjacent pump pulses is 200 μ s at 50 kHz, which is longer than the time

scale of the thermal dissipation process. Consistently, in other reports the repetition rate was usually chosen as above 100 kHz (Nature **462**, 7275 (2009); Nature Photonics **11**, 425 (2017)) for the PINEM experiments, much higher than ours. If a cumulative heating is present, the data points before time zero tend to show some trend, whereas ours don't. Therefore, we believe that the undesirable thermal heating effect such as thermal accumulation is negligible in our experiments.

Comments #5:

The V(IV) valence state disproportionate to V(III) and V(V). Control can be done by XPS technique. Please, add results of the analysis of the VO₂ sample before and after experiments.

Response:

We agree with the referee that using XPS technique one can determine very well the valence state of a material. However, XPS is usually applied to bulk materials or for an ensemble of nanoscale objects where it only provides an orientation-averaged measurement. Due to the probe size and yield efficiency for single NWs, such as those studied in this work, XPS would be extremely challenging, if not impossible. On the other hand, EELS can provide the necessary single-particle sensitivity and measurements of individual VO₂ NWs before and after experiments are indeed possible. Here, we have measured the EELS spectrum of a VO₂ NW before and after the repetitive laser irradiation at 400 nm (similar fluence and repetition rate as that used in our pump-probe experiment) for 10 s. As shown in Fig. R1 below, the EELS signal of the V-L-edge fine structure (L3/L2 ratio) of the NW before and after laser irradiation shows no detectable change. Therefore, under our experimental condition, we can exclude irreversible effects induced by the laser illumination.

Figure R1. EELS results of a VO₂ NW before and after laser irradiation

Reviewer 2

Comments #1:

The authors introduce a novel technique for ultrafast near-field electron microscopy experiments on a single VO₂ nanowire to probe the photoinduced metal-insulator transition in a single nanostructure. As mentioned in the paper, this photoinduced transition has been extensively studied over the years through time-resolved electron scattering, probing structural dynamics, and ultrafast optical spectroscopies which typically measure the spatially averaged electronic response. Here the authors apply PINEM, photo-induced near-field electron microscopy, to measure the dynamics of the local dielectric response of VO₂ following photoexcitation sufficient to induce an insulator to metal phase transition (IMT). In PINEM, electrons do not actually pass through the material but instead sense the local fields surrounding the structure via their contributions to the inelastic free-free transitions of an electron in a photon field.

The novelty of this work lies in the introduction of a second optical pulse to photoexcite the sample at a different wavelength and crossed polarization so that the interaction can be approximated as arising from two independent fields. Thus one can induce a material change with the first pulse and use the second optical pulse to gate the temporal response of the electron-matter interaction. In this way, the time-resolution of the technique can be less than the electron pulse duration.

In my opinion, this is really nice work showing the power of such a pump-probe PINEM technique to probe insulator to metal transitions in correlated electron materials, of which VO₂ is the quintessential example. I find the results of these experiments compelling and demonstrate that it is possible to spatially probe the dynamics of the photo-induced transition. This technique will undoubtedly be useful to study a variety of correlated electron systems yielding a photoinduced IMT where the local dielectric environment is drastically changed due to optical perturbation. In terms of learning new things about the material, I am not sure we really learn something new about VO₂ and they restrict their measurements to the low carrier density regime where the transition is of typical Mott character. It would have been interesting to drive the transition further above the threshold where a quasi-stable structural transition occurs from a monoclinic insulator to a rutile metal.

My first question is: why did the authors not do this? Surely they had enough optical fluence to push the system into the rutile phase. How close were they in doing this, by their estimates? The answer to this may be important in understanding the limitations of this technique. Perhaps the structural transition

does not yield a sufficient dielectric change compared to the contributions of the mobile carriers. In this case, I am not certain I would agree that PINEM could be used to simultaneously probe structural dynamics.

Response:

We greatly appreciate the referee's careful review and high evaluation of our work.

The pump fluence used in our two-color PINEM measurement is $\sim 15.3 \text{ mJ/cm}^2$, which is much larger than the optical gating pulse ($\sim 4.1 \text{ mJ/cm}^2$). As we will demonstrate below in the reply to comment #6, the optical energy density injected in the VO_2 NW by the 400 nm pump beam is 3.5 eV/nm^3 , which is larger than the critical dose of 2 eV/nm^3 . Therefore, we have reliably reached the phase transition. Although our laser system can provide a higher laser fluence and we have actually tried higher pump fluences, a higher pump fluence, however would induce more damage to the thin Si_3N_4 membrane (20 nm) on which the nanowire is sitting, as well as to the nanowire itself.

Finally, we briefly address the possibility to simultaneously probe both electronic and structural dynamics by the two-color PINEM approach. In our work, we showed that both the intensity and the spatial distribution of the PINEM signal substantially change at the equilibrium rutile metallic phase compared to that of the equilibrium monoclinic insulating phase. This indicates that the structural transition does yield a sufficient dielectric change that can be distinguished and probed by PINEM. Therefore, when temporally-gated PINEM electrons are used to perform ultrafast dark-field imaging or ultrafast diffraction, it would be indeed possible to simultaneously capture the structural dynamics as well.

Comments #2:

If the authors have done a fluence dependence, does the long time constant they observe decrease with increasing fluence as reported in other studies?

Response:

We thank the referee for this comment. The main purpose of this work is to demonstrate the possibility of capturing the ultrafast electronic dynamics of individual nanostructures in real space and time by the two-color PINEM approach, and we mainly focused on the emphasis of the technique novelty and on the first demonstration of its capabilities. Therefore, we did not carry out measurements on the pump

fluence dependence in this work. This is also related to the current sample configuration, where the VO₂ NW is sitting on a thin Si₃N₄ membrane, which considerably reduces the window of laser fluence for which we could obtain a detectable signal without breaking the thin membrane. We thank the referee for this suggestion. A detailed laser fluence dependence is planned in our follow-up work on this topic in the near future using an improved sample configuration.

Comments #3:

The pump energy is higher in this study (3.1 eV) than the gating field (1.55 eV). For 800 nm excitation, one would expect an instantaneous metallization of Mott character followed by a rapid recovery of the insulating phase on a sub-picosecond time scale. Did the authors think to switch the role of the pump and gating pulses to verify this, or was there a technical reason why this could not be done?

Response:

We thank the referee for these comments. For the two-color PINEM approach and experiments demonstrated in this work, two pre-conditions need to be satisfied: (1) the pump optical pulse P2 driving the Mott transition should not produce any appreciable near-field (i.e., PINEM signal), and (2) the optical gating pulse P1 must have sufficient fluence to produce PINEM with an intense near-field signal but be below the threshold to trigger the transition or dynamics. Therefore, in choosing the wavelengths for both pump and gating pulses, one has to take into account also the excess energy of the photoexcited carriers and the near-field cross section. In the case of VO₂ nanowires, 400-nm light has a larger excess energy than 800-nm, while producing a weaker evanescent near field as evidenced by the calculations in Fig. R2. The latter is generated using the MatScat^{53,54} routine, with which it is possible to evaluate the near-field produced at both wavelengths for an infinite cylinder (assuming the same polarization, orthogonal to the NW, e.g. along y axis). The results show that under the same incident fluence and polarization, the near-field is more pronounced for 800-nm illumination with respect to the 400-nm case.

Figure R2. Near-field (square modulus of the electric field E_z) calculated for illumination with light at 400 nm (a) and 800 nm (b). The NW has a diameter of 300 nm.

As a consequence, in our experiments we have used the 400 nm optical pulse with a high fluence ($\sim 15.3 \text{ mJ/cm}^2$) and with its polarization parallel to the NW axis to trigger the IMT while minimizing any appreciable PINEM signal; whereas we used the 800 nm optical pulse with a relative small fluence ($\sim 4.1 \text{ mJ/cm}^2$) and a polarization perpendicular to the NW axis as optical gating pulse in order to maximize the PINEM excitation but be below the threshold to trigger the transition. So switching pump and gating pulses, despite being technically feasible, will most likely not provide a clear and well defined separation between PINEM signal and photodoping based on our current configuration.

Comments #4:

Since the readout is mediated through the dielectric response of a nanostructure, and the analysis of this requires numerical modelling of the local near-field, it is necessary that the real and imaginary parts of the dielectric function are input for the gating field. The authors quote Ref. 43 however this work focuses on thin films of VO₂ which through the transition are composed of domains of conducting and insulating phases. How sure are the authors that the same dielectric functions appropriate for a film are applicable to a nanostructure where it is likely to be composed of one or two domains?

Response:

The nanowires investigated in our work have diameters of ~ 300 nanometers and extend for distances of several microns, which are far greater than the Fermi wavelengths and exciton Bohr radius in these types of materials. Therefore, any quantum size effect on the electronic properties is negligible. In other words, the nanowires can be considered as ideal bulk samples with flexibility and free of the micro-cracks effect during the phase transition, their dielectric properties do not significantly differ from those of the bulk, or high-quality thin films, especially for optical wavelengths in the visible range as adopted in our experiment (see for instance, Fu et al. J. Appl. Phys. **113**, 043707 (2013)).

We have verified that the basic parameters of our numerical model, such as the absorption cross sections, only change slightly (up to a few percent) when values of the dielectric function for bulk or thin films are considered. Despite such minimal changes, what nevertheless matters for our interpretation of the transition dynamics is not the absolute value of the dielectric function, but rather its relative change across the phase transition, whose trend is also very similar between bulk and thin films.

Finally, we want to point out that the results of our simulations are in good semi-quantitative agreement with the experimental data, and therefore we believe that it is reasonable to consider for our estimation the values of dielectric function reported for nanoscale thin films.

We have included such considerations in the revised version of the manuscript (page 9), as also presented below.

“The transition is modelled as a variation of the dielectric function from $\epsilon_{ins} = 5.68 - i3.59$ ($\tilde{n}_{ins} = 2.49 + i0.72$) in the insulating phase to $\epsilon_{met} = 2.38 - i3.26$ ($\tilde{n}_{met} = 1.79 + i0.91$) for the metallic phase as derived from Ref. 43 for the case of a thin film. In our experiment we consider that the dielectric function of a thin film might represent a good approximation of the dielectric environment of our VO₂ NWs, whose dielectric properties do not significantly differ from those of the bulk or high-quality thin films, especially for optical wavelengths in the visible range as adopted in our experiment⁴⁴.”

Comments #5:

Are plasmonic effects negligible or are they taken into account through the finite element simulations?

Response: All plasmonic effects for the investigated VO₂ nanowires are intrinsically taken into account within the finite element simulations (see also page 9 in the revised manuscript).

“Also, all plasmonic effects for the investigated VO₂ nanowires are intrinsically taken into account within the finite element simulations.”

Comments #6:

The calculation of the carrier density in Equation 6 is likely too simplistic, given it uses things like a reflection coefficient for a nanostructure. Perhaps a scattering model is more appropriate to estimate the carrier density. An accurate number for the carrier density would be helpful to evaluate whether they are truly close to the structural transition.

Response:

We agree with the reviewer’s comment and we thank her/him for pushing us to further investigating this aspect. Following her/his suggestion, we have resorted to scattering theory to determine the absorption cross section for an infinite cylinder (see page 11 (main text) and page 17 (Methods) of the revised manuscript), as shown below.

Main text on page 11:

“In VO₂, a density-driven photoinduced IMT has been proven to occur with a critical energy dose ΔH_C of 2 eV/nm³⁴⁵. For the case of a NW, the optical energy densities injected by the P1 and P2 pulses can be evaluated by resorting to scattering theory. In such framework we determined the absorption cross section, C_{abs} , for the case of an infinite cylinder (see Methods). The evaluated cross section is 2.63×10^{-7} m in the case of 400 nm (P2), while at 800 nm (P1) C_{abs} is 3.06×10^{-7} m. We then computed the absorbed optical energy density (energy per unit volume) injected within the NW as:

$$\rho_{E,ph} = \frac{C_{abs}}{\pi a^2} \phi_{inc} \quad (6)$$

where ϕ_{inc} is the incident fluence and a is the cylinder radius. By considering the experimentally adopted parameters and geometrical configurations, we found that $\rho_{E,ph}$ is about 1 eV/nm³ at 800 nm, while increases to 3.5 eV/nm³ at 400 nm. The fact that $\rho_{E,ph}$ for 800 nm excitation is well below the critical energy dose ΔH_C , whereas $\rho_{E,ph}$ for 400 nm is well above, is a strong confirmation that P2 is able to trigger the ultrafast IMT while P1 acts only as PINEM probe. Note that, electron beam may also

induce effects on the IMT in VO₂, such as lowering the IMT temperature by creating oxygen vacancies⁴⁶. However, the dose of the electron pulse in our experiment is several orders of magnitude smaller than the conventional thermal electron beam and its effect is negligible.”

Methods section on page 17:

“Scattering theory for an infinite cylinder

To determine the absorption cross section for an infinite cylinder we have resorted to scattering theory. In particular, we have used the derivation provided by Bohren & Huffman⁴⁷, where the scattering and extinction efficiencies in case of parallel (I) polarization (used for P2) are given by:

$$Q_{sca,I} = \frac{P_{abs,I}}{2aL\phi_{inc}} = \frac{2}{x} [|b_{0I}|^2 + 2 \sum_{n=1}^{+\infty} (|b_{nI}|^2 + |a_{nI}|^2)] \quad (7a)$$

$$Q_{ext,I} = \frac{P_{ext,I}}{2aL\phi_{inc}} = \frac{2}{x} Re\{b_{0I} + 2 \sum_{n=1}^{+\infty} b_{nI}\} \quad (7b)$$

whereas in the case of normal (II) polarization (used for P1) we have:

$$Q_{sca,II} = \frac{P_{abs,II}}{2aL\phi_{inc}} = \frac{2}{x} [|a_{0II}|^2 + 2 \sum_{n=1}^{+\infty} (|a_{nII}|^2 + |b_{nII}|^2)] \quad (8a)$$

$$Q_{ext,II} = \frac{P_{ext,II}}{2aL\phi_{inc}} = \frac{2}{x} Re\{a_{0II} + 2 \sum_{n=1}^{+\infty} a_{nII}\} \quad (8b)$$

Thus, the absorption efficiencies can be evaluated as:

$$Q_{abs,I} = Q_{ext,I} - Q_{sca,I}, \quad Q_{abs,II} = Q_{ext,II} - Q_{sca,II} \quad (9)$$

where a is the NW radius, L its length, a_n and b_n are coefficients that depend on the Hankel and Bessel functions, as reported in Bohren & Huffman⁵². The absorption cross section per unit length C_{abs} is related to the corresponding efficiency via the geometrical factor $2a$ as $C_{abs} = 2a Q_{abs}$. The evaluated cross section C_{abs} is 2.63×10^{-7} m in the case of 400nm, while at 800 nm C_{abs} is 3.06×10^{-7} m. We then computed the absorbed optical energy density (energy per unit volume) injected within the nanowire as:

$$\rho_{E,ph} = \frac{c_{abs}}{\pi a^2} \phi_{inc} \quad (10)$$

where ϕ_{inc} is the incident laser fluence. The numerical evaluation has been implemented using two different codes: MatScat^{53,54} and ICOTOOL⁵⁵, which provided identical results. Here, we have considered $\tilde{n}_{ins} = 2.49 + i0.72$ at 800 nm and $\tilde{n}_{ins} = 2.09 + i1.31$ at 400 nm. By considering the experimentally adopted parameters and geometrical configuration, we found that the energy density injected at 800 nm is about 1 eV/nm³, while at 400 nm we have about 3.5 eV/nm³. The fact that $\rho_{E,ph}$ for 800 nm excitation is well below the critical energy dose ΔH_C for a photoinduced IMT phase transition of 2 eV/nm³, whereas $\rho_{E,ph}$ for 400 nm is well above, is a strong confirmation that only the blue light pulse will be able to drive the transition.”

To be coherent throughout the manuscript we have also modified the integral equation (Eq. 8) in the main manuscript, expressing the absorbed (excessive) thermal energy density $\rho_{E,th}$ for the VO₂ NW as:

$$\rho_{E,th} = \frac{c_{abs}}{\pi a^2} \phi_{inc} \left(\frac{h\nu - E_g}{h\nu} \right) = \int_{T_0}^{T_0 + \Delta T} C_{ph}(T) dT$$

where $E_{g_{VO_2}} = 0.7$ eV is the energy band gap as indicated in Ref. [Fu, Deyi, et al. "Comprehensive study of the metal-insulator transition in pulsed laser deposited epitaxial VO₂ thin films." *Journal of Applied Physics* 113.4 (2013): 043707]. The induced temperature jump is about 31 K, which is very similar to the value determined before (~ 26 K) and well below the transition temperature of 68 K. Because of such slightly different estimate, we have modified the horizontal axis T_{eff} in Fig. 2c (page 30 in the revision) and Fig. R3 below. For the Si₃N₄ membrane, Eq. 7 is still correct and remains unchanged.

Figure R3. Revised version of Figure 2b-c

Comments #7:

On page 13, the authors assert that they see no evidence for an intermediate state on traversing the insulating to metallic phase. They indicate that previous works provide evidence for such an intermediate state, and they cite the work of Baum (Ref. 9) and Siwick (Ref. 10). While Baum proposed a two-step approach to the transition in 2007, the Siwick group to my knowledge has never made this assertion and in fact their data disproved it in 2014. They describe the transition as a disordering transition due to the non-thermal melting of the V-V dimerization in a fluence-dependent fraction of crystallites. They are quoted in their Ref. 10 work: “Thus, the fast dynamics correspond to nonthermal melting of the PLD in a fluence-dependent fraction of crystallites. In these crystallites, the vanadium atomic positions relax to their equilibrium R-phase separation on a time scale of 300 fs (26).” This was further explored in a recent PNAS paper (M. R. Otto et al., 116, 450 (2019)) in a percolative picture of the transition over the film, so this appears to be fairly consistent in their interpretation. No soft-mode phonons indicating an intermediate was seen in either work. This mistake can be forgiven as it appears to be a misinterpretation of the Siwick by the Wall Science paper (Ref. 7 in this manuscript) who appear to cite Siwick and Baum together concerning the intermediate. This should be likely be corrected in the literature but at the very least should not be propagated.

Overall, I believe this work to be highly interesting to the community and can provide useful information concerning correlated electron physics in nanostructures. I support publication of this work provided answers to some of the questions I've asked are reasonably answered and corrections are made.

Response:

We greatly appreciate the referee's valuable comments on the discussion of the mechanism and supporting the publication of our work after corrections and revision. We agree with the referee's comments. According to her/his suggestion, we have deleted in the revised manuscript (page 13) the inappropriate assertion and citations about the intermediate states in the IMT of VO₂ to avoid any misleading interpretation, as also reported below.

“As shown in Fig. 4b, and Fig. 4d, the localized PINEM signal from the VO₂ NW observed in our experiments exhibits an initial ultrafast dynamical process with a time constant of ~150 fs followed by a slower recovery process, which indicates the transition from the insulating to metallic phase without evidence of passing through any intermediate states.”

Comments #8:

Minor questions/comments:

In Fig. 4b there is a blue box drawn, however there is no obvious indication as to what it is.

For Fig. 4e, the fit to the width is allowed to go negative. What does this mean?

Response:

The blue box in the right panel of Fig. 4a show the area along the NW where the PINEM signal (presented in Fig. 4b-e) is integrated, which has been mentioned in page 12 of the main text. To make it clearer, we have added a sentence in the figure caption in the revised manuscript (page 35), as shown below:

“The blue box in the middle panel indicates the area along the NW where the PINEM signal is integrated.”

Regarding to Fig. 4e, we have now corrected the best fit to the data with a model that does not allow the curve to become negative (page 34 of the revised manuscript), as shown below.

Revised version of Figure 4e

Reviewer 3

Comments:

The Fu et.al. manuscript is presenting the ultrafast dynamics of insulator-metal phase transition of single VO₂ nanowire by probing the photoinduced dielectric constant using the photon gating approach in the ultrafast electron microscope. This approach has been utilized to study the phase transition in VO₂ nanoparticles previously in ref#31 in the manuscript. The author should cite the previously mentioned reference in the introduction (i.e. line 54 & line 61). Although, the novelty of the presented work is the connection between the ultrafast dynamics and the morphology of single VO₂ nanowire which consider an important step forward in this approach and adding an impact in the field to study and image the ultrafast dynamics of solid-state by the ultrafast electron microscope.

Response:

We thank the reviewer for the careful review, positive comments and constructive suggestions. Following the referee's suggestion, we have cited the mentioned references in the introduction of the revised manuscript (page 3), as also reported below:

“Recently, a photon gating approach in transmission electron microscopy has been proposed for studying the phase transition of VO₂ nanostructures³¹, where the authors have investigated the spatially-average IMT dynamics of an ensemble of VO₂ nanoparticles.”

Comments #1:

The organization of the experiments and starting with the one-color PINEM experiment first to prove the effect of the phase transition on the PINEM spectrum is well presented.

The two-color PINEM experiment explanation and satisfaction of the two mandatory preconditions of this type of experiment, namely; 1- The pump optical pulse should not produce any PINEM signal, and 2- The optical gating pulse must be below the threshold to trigger the dynamics under study, reflects the capability and understanding of the photon-gating PINEM experiment.

However, I think the authors miss present the satisfactory of condition #2 in figure 2a (as the author refers to in line 222) where the middle and the left panel show the PINEM signal of P1 and I think the author meant to present in the right panel that no obvious PINEM signal is generated from P2 pulse.

Response:

We thank the reviewer for these comments. In Fig. 2a, the left panel shows the bright field image of the VO₂ nanowire and the middle panel shows the PINEM image of the P1 gating pulse (wavelength of 800 nm) with the polarization perpendicular to the nanowire axis (maximize the local field excitation) with a relative small fluence of $\sim 4.1 \text{ mJ/cm}^2$, where strong PINEM signal is visible around the nanowire. Under such a relative small fluence, the 800 nm gating pulse can only induce a temperature jump of $\sim 31 \text{ K}$ (page 8 and Methods in page 19) for the nanowire. Therefore, the condition #2 “The optical gating pulse must be below the threshold to trigger the dynamics under study” is satisfied. The right panel of Fig. 2a shows the PINEM image of the P1 gating pulse (wavelength of 800 nm) with the polarization parallel to the nanowire axis (minimizing the local field excitation) with the same fluence, where no obvious PINEM signal is generated around the wire except at the end. This comparison is to demonstrate that a polarization of the optical field parallel to the nanowire axis will not produce any appreciable PINEM signal. Thus, we set the polarization of the P2 pump pulse parallel to the nanowire axis in the two color PINEM experiment to satisfy the condition #1: “The pump optical pulse should not produce any PINEM signal.”

We understand that at this point along with the right panel of Fig. 2a the manuscript might appear misleading, and therefore we have decided to remove it from the revised version of the figure. Instead, we have added in the revised version of Fig. 4 (see page 34 of the revision and also below in the reply to comment #3) an additional panel in which we report PINEM measurement on the nanowire with only P2 pump pulse excitation (wavelength of 400 nm, fluence of 15.3 mJ/cm^2 , above the IMT threshold) at $t = 0 \text{ ps}$ and its polarization parallel to the nanowire axis. Indeed, no apparent PINEM signal was observed around the wire. Therefore, both the two mandatory pre-conditions are satisfied in this two color PINEM experiment.

Comments #2:

The author presented the change in the PINEM spectra due to the photoinduced phase transition and the change in the dielectric of VO₂ nanowire in Figure 4 a and then show the normalized spectra in Figure 4b. I do not think this necessary and it would confuse the reader. It should be discarded or move to supplementary information. The same applied to Figure 4 e.

Response:

We thank the reviewer for this comment. In Fig. 4b-c, we present the PINEM intensity change across the phase transition, which reflects the change of the dielectric function of the nanowire and thus the IMT dynamics. In Fig. 4d-e, we present the normalized spectra in order to better show the change of the full width at half maximum of the PINEM signal profile, i.e. the spatial distribution change of the localized field across the IMT. As discussed in the main text (page 12), the change of the dielectric function of the nanowire across the IMT will also result in changing the spatial distribution of the localized field and thus that of the PINEM signal. Both of them reflect the electronic dynamics in the IMT of the VO₂ nanowire and they are consistent with each other in our experiment. Furthermore, the presented time-dependent nanoscale variation of the spatial distribution of the PINEM signal demonstrates the strong capability of this approach for capturing the ultrafast phase transition dynamics at nanometer scale in real space. Therefore, we believe that by showing both the measured PINEM profiles and their normalized versions, it would better emphasize the versatility of information that one could retrieve with the two-color PINEM technique.

Comments #3:

It is important to improve the presentation quality of the results shown in figure 4d to show the error bar of the data and mention the numbered of the scans that have been recorded in the text.

Response:

As suggested by the reviewer we have added the error bars in Figures 4c and 4e. The error bars have been obtained by considering the measurement uncertainty and the variance within the counts. In our experiments, we have recorded four scans in imaging mode and two scans in spectroscopy mode. They are all showing very similar results and therefore the data presented in the manuscript, which has been obtained by properly averaging the near field around the nanowire, are representative of the true behavior. We have included such information in the Methods section of the revised manuscript (page 34 and page 35). Below we include the revised version of Fig. 4.

Revised version of Figure 4

Comments #4:

Finally, in the conclusion section, the authors mentioned that the photon gating approach in electron microscopy would lead to achieving the attosecond resolution in electron microscopy by using the advances of in attosecond optical pulse generation then cite ref# 55& 56. Obviously, the authors are not from the attosecond physics field, and this probably why they cited these two references referring to optical attosecond pulses. These two citations reporting the advancement of XUV attosecond pulses

which is totally different than optical attosecond pulse. Using the term “optical” means in the visible and nearby frequency ranges not extreme ultraviolet (i.e. XUV). The generation of PINEM inside the microscope using attosecond XUV pulses is technically very changeable due to the limitation of performing the HHG process (the only valid way so far in the field. to generate XUV attosecond pulses) inside the microscope. Moreover, even if this limitation is overcome (for example by considering the HHG process in solid), the XUV generation efficiency and flux are not sufficient to generate any PINEM signal. The PINEM-like experiment is reported in the attosecond physics field for free electrons two decades ago and it is called attosecond XUV streaking. Hence, in line (326) in the manuscript conclusion, citing reference#32 in the manuscript and [J. Phys. B: At. Mol. Opt. Phys. 51, 032005 (2018)] will be more accurate and realistic.

Considering the mentioned points is crucial for improving the work presentation. The manuscript figures (i.e. Fig 2 and Fig 4), language, and citations need to be corrected and revised.

Response:

We thank the referee for the careful review and for providing the frontier of generation of attosecond pulses. Following his suggestion, we have replaced the two previous references with the two mentioned by the reviewer in the conclusion part of the revised manuscript.

REVIEWERS' COMMENTS

Reviewer #2 (Remarks to the Author):

The authors have addressed all comments and concerns and I believe have significantly improved the manuscript. Certainly in the treatment of the total pump excitation, the scattering model seems more applicable. I support publication in the manuscript's current form.

Reviewer #3 (Remarks to the Author):

The author answered all the raised questions and improved the manuscript accordingly. However, I have one more comment. In the revised version

In the revised version line 65-67;

"Recently, a photon gating approach in transmission electron microscopy has been proposed for studying the phase transition of VO₂ nanostructures¹⁹, where the authors have investigated the spatially- average IMT dynamics of an ensemble of VO₂ nanoparticles."

Unfortunately, the author cites ref 19 without reading it carefully despite the similarity with the reported work in this manuscript. Ref 19 demonstrated the measured of the ultrafast dynamics in VO₂ nanostructure, not just propose it. I do not think admitting it will affect the presented manuscript's credentials.

Response to reviewers' comments

We would like to thank the reviewers for their positive and constructive reviews, and supporting for publication of our work. We have addressed the additional comments in this revised version. Our detailed point-by-point responses are listed below.

Reviewer #2 (Remarks to the Author):

Comments:

The authors have addressed all comments and concerns and I believe have significantly improved the manuscript. Certainly in the treatment of the total pump excitation, the scattering model seems more applicable. I support publication in the manuscript's current form.

Response:

We thank the referee for the careful review and supporting publication of our work in the current form.

Reviewer #3 (Remarks to the Author):

The author answered all the raised questions and improved the manuscript accordingly. However, I have one more comment. In the revised version in the revised version line 65-67;

“Recently, a photon gating approach in transmission electron microscopy has been proposed for studying the phase transition of VO₂ nanostructures¹⁹, where the authors have investigated the spatially-average IMT dynamics of an ensemble of VO₂ nanoparticles.”

Unfortunately, the author cites ref 19 without reading it carefully despite the similarity with the reported work in this manuscript. Ref 19 demonstrated the measured of the ultrafast dynamics in VO₂ nanostructure, not just propose it. I do not think admitting it will affect the presented manuscript's credentials.

Response:

We thank the referee for this further comment. We have revised the description on the citation of Ref. 19 according to the referee's suggestion in the revised manuscript (page 3), as also shown below.

“Recently, a photon gating approach in transmission electron microscopy has been demonstrated for studying the phase transition of VO₂ nanostructures¹⁹, where the authors have investigated the spatially-average IMT dynamics of an ensemble of VO₂ nanoparticles.”